



# Climate change and runoff contribution by hydrological zones of cryosphere catchment of Indus River, Pakistan

Kashif Jamal[1,2], Shakil Ahmad[4], Xin Li[1,3], Muhammad Rizwan[1,2], Hongyi Li[1,2], Jiaojiao Feng[1,2]

[1]Key Laboratory of Remote Sensing and Geospatial Science, Northwest Institute of Eco-Environment and Resources, Chinese
Academy of Sciences, Lanzhou, 730000, China
[2]University of Chinese Academy of Sciences, Beijing, 100049, China
[3]CAS Center for Excellence in Tibetan Plateau Earth Sciences, Beijing, 100101, China
[4]School of Civil and Environmental Engineering, National University of Sciences and Technology, Islamabad, 44000, Pakistan

*Corresponding to:* Xin Li (lixin@lzb.ac.cn)

**Abstract.** Climate change has significant impacts on hydrology in high altitude snow and glacier covered mountainous regions. These regions are highly sensitive to changes in climate variables, such as temperature and precipitation and producing high runoffs. Runoff produced from different altitude ranges and their sensitivity to current and changing climate is also unknown. This study was carried out in high altitude mountainous cryosphere Hunza River Catchment (HRC) which is located in Hindukush and Karakoram ranges and is the major tributary of the Indus River Basin. Snowmelt-Runoff Model (SRM) was used to analyse the current and projected hydrological regimes and the sensitivity of Snow Cover Area (SCA) at different altitude levels under current and changing climate. Under the current condition (i.e., 2001–2010 except 2006), the results showed that about half of the mean annual streamflows at the outlet of the HRC is contributed by the altitude ranges of 4500–5500 m a.s.l. Climatic projections under the RCP8.5 and RCP4.5 scenarios were used for the climate change impact assessment. Compared to the baseline climate, the mean annual temperature would increase by 0.7 (0.6), 2.4 (1.3) and 4.6 (1.9) °C, respectively during 2030s, 2060s and 2090s; and the mean annual precipitation would increase by 63.3 (33.6) mm during 2090s under the RCP8.5 (RCP4.5) projections. Moreover, two SCA scenarios were developed, i.e., the baseline unchanged SCA and the hypothetical change in SCA scenarios. In the first SCA scenario, the results showed that additional streamflows of 43 (34), 153 (83.4) and 304 (115.7) $m^3 s^{-1}$ under RCP8.5 (RCP4.5) will be added into baseline annual streamflows of 269 $m^3 s^{-1}$ during 2030s, 2060s and 2090s, respectively. In the second scenario, we found that 10 % and 15 % decrease in SCA would result in increases (or decrease) in streamflows approximately by 18 (2) % and 42 (7) % under the RCP8.5 (RCP4.5) scenario during 2060s and 2090s, respectively. Whereas altitude range 4500–5500 m a.s.l showed increasing trend during pre-monsoon (April–June) and monsoon (July–August) season under changed SCA scenario for both RCPs scenarios. Current and near future climate pattern is favourable for Indus River regarding high water flows. However, future water flow pattern is declining because of disappearance or decrease in snow and glaciers melt area which correspondingly means that mid/downstream water allocation will be effected or reduced at some extent. Proper adaptations or managements strategies should be executed for upcoming harsh conditions.



## 1 Introduction

The hydrological processes in cryospheric mountainous catchments play a very important role in snow and glacier melt to stabilize the river flows, resulting in a very low inter-annual variation (Li et al., 2018). For that reason, mountainous area that are commonly engaged in high precipitation rate, snow melt and glaciers melt (Cheng et al., 2014). Hydrological modelling is important for good water resources management in the perspective of floods, droughts and irrigation purposes under projected climate which is relatively thought-provoking due to the ambiguity in climate change projections (Immerzeel et al., 2011). Climate change has significant impacts on snow cover area (SCA) in mid-latitude mountainous regions. Spatiotemporal changes in SCA and the runoff produced are the indicator of climate change (Wang and Li, 2006). The impacts of the projected climate on streamflows are prophesied by the employment of hydrological models that are strictly associated with the simulation efficiency of the models (Azmat et al., 2016a). However, the application of an suitable hydrological model in high-altitude cryospheric (snow and ice) catchments is essential as of two streamflows sources i.e. rainfall-runoff and snow- and glacier-melt runoff (Azmat et al., 2015).

The performance of the hydrological models is challenged in high-altitude regions because of the enormous involvements of snow- and glacier-melt runoff (Martinec et al., 2008). Some processes like energy and mass balance; mass and heat transport by sublimation and vaporization make the hydrological model less efficient (Tarboton and Luce, 1996). Nowadays, numerous hydrological models by means of both snowmelt and rainfall runoff processes have been developed to model the streamflows of a catchment that is categorized by snowmelt and rainfall runoff contribution (Ohara et al., 2010; Şensoy, 2005). Some rainfall-runoff models, also have capability up to some extent to model the snowmelt runoff process were observed not as much of effective in high-altitude catchments (Azmat et al., 2016a). Therefore, the modelling of streamflows in high-altitude catchments is all the time a challenge for researchers to model the streamflows accurately.

Previously, some studies have been carried out to simulate daily streamflows of high-altitude snow and glacierized catchments. Several researchers (Rulin et al., 2008; Tahir et al., 2011b; Kult et al., 2014; Romshoo et al., 2015; Azmat et al., 2016a) have used SRM with integration of SCA product of MODIS satellite and found that SRM have capability to model the streamflows of high altitude catchments.

Further, climate change impact on streamflows required future climate data at some spatial and temporal resolution. Therefore, GCMs (General Circulation Models) are the unique sources for projected climate variables. These climate models have course spatial resolution of climate variables and the projected variables can be directly used for regional climate change impact studies (Berg et al., 2012; Piani et al., 2010). In addition, all climate models comprises excessive biases in their outputs that needs to be corrected first otherwise may pose lot of inaccuracies in climate change studies. Therefore, it is necessary to bias correct the GCMs' outputs prior to their application in hydrological modelling (Terink et al., 2009; Teutschbein and Seibert, 2012; Burhan et al., 2015).

Moreover, Kabiri et al. (2015) applied HEC-HMS and climate data derived from GCMs by using statistical downscaling model (SDSM) to study the climate change impact on hydrological response of Klang Watershed, Malaysia. Singh et al. (2015) used



statistical downscaling technique to generate future time series of temperature for North-Western Himalayan region of India. He stated that projected increase in temperature will alter the time and length streamflows in major rivers of Himalayan region. Kazmi et al. (2014) used statistical downscaling technique to generate projected temperature for Indus Basin of Pakistan and increasing trends for temperatures mainly in the northern and southwestern areas was observed.

Several researchers (Wilby et al., 2008; Goulden et al., 2009; Allamano et al., 2009) carried out hydrological modelling by using GCMs' output directly to investigate the projected climate change impact water resources. Mohammed et al. (2015) utilized the CMIP5 (Coupled Model Inter comparison Project phase 5) climate data for Representative Concentration Path ways (RCPs) i.e., RCP8.5 and RCP4.5 with incorporation of RHESSys (Regional Hydro-Ecological Simulation System) model to examine the hydrological regime of Champlain Basin. Further, Piras et al. (2016) used statistical downscaling techniques

with integration of GCMs and RCMs to analyse the climate change impact on precipitation and extreme runoff events by using tRIBS (TIN-based Real-time Integrated Basin Simulator) hydrologic model in a Mediterranean basin.

In this study SRM was used to simulate streamflows volume generated at different altitude zones of HRC in past and future time steps in conjunction with MODIS snow cover dataset. Indus Gang and Brahmaputra (IGB) climate dataset based on future emission scenarios of the Representative Concentration Pathways, RCP4.5 and RCP8.5 with 4.5 W m$^{-2}$ and 8.5 W m$^{-2}$ radiative

forcing by 2100, respectively (Ali et al., 2015; Burhan et al., 2015) was utilized which is available at fine spatial resolution of 5 km for future streamflows projections. The main objective of this study is to investigate the contribution of streamflows from different altitude zones and to find the dominant altitude range which is more sensitive to current and changing climate. Previous studies (Rulin et al., 2008; Tahir et al., 2011b; Romshoo et al., 2015; Azmat et al., 2016a) applied SRM on whole the catchment and described the contribution of streamflows anomalies over the catchment.  This study used a simple approach to

capture the deficiencies present in the previous studies to give a better understanding for future water management.

## 2 Study Area

This study was conducted in HRC having catchment area approximately 13 718 km$^2$ (Fig. 1a), located in high-altitude (ranges from 1395 to 7849 m a.s.l., see Table 1) mountainous region of central Karakoram, in the north of Pakistan. Approximately, 32.5 % (4460 km$^2$) of the total catchment area is located above 5000 m a.s.l. which is considered as perennial glacier part of

the catchment (Akhtar et al., 2008). The highest altitude is 7849 m a.s.l. which is located at Distaghil Sar in Hispar Muztagh sub-range of Karakoram range, while the lowest altitude is 1395 m a.s.l at Dainyor Bridge. The hydrological regime of Hunza River is mainly fed by seasonal snow and glacier melts with little influence of summer monsoon, while winter precipitation due to westerlies plays an important role in snow accumulation and nourishment of glaciers (Dahri et al., 2016). However, high altitude precipitation (i.e. snow fall) behaviour is still unknown as well as quite uncertain.

**Figure 1a: Location map of Hunza River Catchment (HRC).**

**Table 1: Salient features of HRC.**



The basin-wide maximum (80 %) and minimum (34 %) SCA varies during winter and summer season, respectively, as confirmed by Tahir et al. (2011a). While, the mean annual precipitation at four climate stations, Hunza (2156 m a.s.l.), Naltar (2858 m a.s.l.), Ziarat (3669 m a.s.l.) and Khunjrab (4730 m a.s.l.) are 389, 679, 247 and 187 mm, respectively, while the mean annual streamflows at Dainyor Bridge is 323 $m^3 s^{-1}$ (Tahir et al., 2011a and b). The strongest hydrological region of the HRC
is above 5000 m a.s.l. where five- to tenfold precipitation increase as reported by Hewitt (2005, 2007) with a large drop in temperature, resulting snow and glacier accumulation.

# 3 Dataset

## 3.1 Topographical data

The Advance Spaceborne Thermal Emission and Reflection Radiometer Global Digital Elevation Model (ASTER GDEM) of
30m spatial resolution was used in this study for the delineation of catchment and extraction of physical parameters (area, elevation, etc.). The study area was divided into six (6) altitude zones with the elevation difference of 1000 m between two successive altitude zones for the zone-wise (ZW) application of SRM (see Fig. 1b and  Table 1). Several key features of different hydro-climate stations, elevation zones are given in Table 1.

**Figure 1b: Hypsometric curve showing elevation zones of HRC**

## 3.2 Hydro-climatic data

In Pakistan, generally, the hydro-meteorological data mostly recorded by the WAPDA (Water and Power Development Authority) and PMD (Pakistan Meteorological Department). The meteorological data (daily precipitation and temperature) during 2001 to 2010 was collected from WAPDA for three climate stations (Naltar, Ziarat and Khunjrab), while, for Hunza
climate station, the data (2007 to 2013) was obtained from PMD. Further, the streamflows data at Dainyor Bridge was obtained from Surface Water Hydrology Project of the WAPDA (SWHP-WAPDA), during 2001–2010 (except 2007 year). The observed hydro-climatic data for a base period of 9 years i.e. 2001 to 2010 (except 2007 year) was used as reference dataset which is henceforth referred as baseline (observed).
Further projected climate data of temperature and precipitation was collected from Himalayan Adaptation, Water and
Resilience (HI–AWARE) project named as IGB climate dataset for Indus, Ganges and Brahmaputra River Basins. This dataset comprise total eight (8) GCM runs (four RCP4.5 and four RCP8.5) of CMIP5 and is available at 5km spatial resolution. The detailed explanation about selection and generation of this dataset is given by Lutz et al. (2016).

## 3.3 Snow cover data

The MODIS (Moderate Resolution Imaging Spectroradiometer) Snow Cover Area (SCA) product MOD10A2 at 8-days
interval is available on Terra satellite with nearly 500m spatial resolution, was selected for the determination of percentage of





SCA over HRC. Total 450 MOD10A2 images, during 10 years (2001 to 2010), were downloaded from http://nsidc.org/cgi-bin/snowi/search.pl. Images were then further projected to WGS1984 UTM 43N ZONE projection system. ArcGIS tool was used for extraction zone-wise SCA for the HRC (Fig. 2). Since, the existence of clouds is always a problem for the extraction of accurate SCA. Therefore, image with cloud cover more that 10 % was rejected and SCA on that day was assessed by interpolation of previous and next available image. This SCA product has been used successfully by several researchers for the streamflows simulations (Tekeli et al., 2005; Tahir et al., 2011a and b; Tahir et al., 2016; Romshoo et al., 2015; Mukhopadhyay and Khan, 2015; Hasson et al., 2014; Azmat et al., 2016a and b). Linear interpolation was used to obtain daily zone wise SCA for further used in SRM hydrological model.

**Figure 2: Zone wise seasonal percent SCA over 10 years (2001–2010) duration for HRC.**

## 4 Methodology

The daily streamflows simulation was carried out at Dainyor Bridge using Snowmelt Runoff model. The model was calibrated for 6 years (2001–2006) and validated for the 3 years (2008 to 2010) durations. Further the SRM was run for the assessment of climate change impact on streamflows using IGB climate datasets of 8 GCMs runs as described in coming Sect. 5.2. The methodology used in this study is described in Fig. 3, below.

**Figure 3: Schematic diagram showing the methodology used for the study of HRC.**

### 4.1 The Snowmelt Runoff Model (SRM)

SRM is a temperature based energy dependent degree-day hydrological model used to simulate daily streamflows in mountainous catchments where snowmelt is a main runoff factor. SRM computes daily runoff produced from rainfall and snowmelt according to the following Eq. (1):

$$Q_{n+1} = Q_n + f(T_n, P_n, S_n) \qquad (1)$$

Where Q, T, P and S are the daily streamflows ($m^3\ s^{-1}$), Temperature (ºC), Precipitation (mm) and Snow Cover Area (%), respectively. The detail description of Eq. (1) is described in SRM user's manual (Martinec et al., 2008).

For the zone wise application of SRM, HRC was divided into six (6) altitude zones with difference of 1000 m elevation as discussed by Tahir et al. (2011b) and Azmat et al. (2016a) are shown in Table 1. The hypsometric analysis shows that 44.7 % of catchment area is located in Zone(4) that elevation ranges from 4501–5500 m a.s.l. Since, the Zone(5) and Zone(6) have no climate station, therefore, mean temperature for both zones were assessed by the extrapolation of observed daily mean temperature using the lapse rate value of 0.7 ºC 100 $m^{-1}$. The zone-wise daily temperature at mean elevation of each zone was estimated by using lapse rate method. The zone wise mean daily (2001–2010) temperature distribution for HRC is shown in Fig. 4.



**Figure 4: Zone-wise distribution of temperature (°C) and snow cover area (%), represented by dash lines and full line, respectively, during 2001–2010 over HRC.**

The precipitation data of four climate stations was considered for four respective zones as described in Table 1 and average precipitation of Zone(3) and Zone(4) was considered for Zone(5) and Zone(6). Similarly, the daily SCA for each zone was produced by the linear interpolation of percent SCA extracted from cloud free MODIS images at 8-days interval. Fig. 4 is showing the zone wise mean daily (2001–2010) SCA over HRC.

The initial DDF value for snow, ice and glaciers were obtained from the previous studies carried out in Himalayan and Karakorum region (Zhang et al., 2005; Hock, 2003). The simulation accuracy of SRM during calibration and validation was measured by using statistical parameters i.e. Nash–Sutcliffe coefficient (NSE) and coefficient of determination ($R^2$).The various parameters optimized for different zones in SRM are given in Table 2.

Once the optimized parameter values were obtained the model was again run for each altitude zone to individually calculate the runoff generated by all the zones. For that purpose, at one instant of time each simulation was limited to only one zone to actively participate in runoff generation and at the same time other five zones were restricted to participate in runoff generation. In this scenario runoff generation for whole catchment was obtained by performing six simulations (one for each for one interval of time).

## 4.2 Climate change impact assessment on streamflows

In this study eight GCM runs [BNU-ESM_r1i1p1, inmcm4_r1i1p1, CMCC-CMS_r1i1p1, CSIRO-Mk2-6-0_r4i1p1 (RCP4.5); inmcm4_r1i1p1, CMCC-CMS_r1i1p1, bcc-csm1-1_r1i1p1, CanESM2_r3i1p1 (RCP8.5)] were scrutinized by Lutz et al. (2016) from 163 GCM runs obtained from Coupled Model Intercomparison Project Phase 5 (CMIP5), on the basis of extreme projections. The projected precipitation and temperature datasets for aforementioned eight (8) General Circulation Models (GCMs) downscaled at 5×5 km grid size were obtained from HI–AWARE project. Further, detailed description of the aforementioned datasets used in current study is given by Lutz et al. (2016).

The climate change impact assessment was carried out on each altitude zone to examine the contribution of streamflows by individual zone on different time steps.  Firstly, the daily precipitation and temperature data of RCPs (8.5 and 4.5) was bias corrected by engaging delta bias correction technique (Sect. 4.3) to study the projected climate changes in each altitude zone of the study area. Secondly, the hypothetical change in each zone's SCA was adopted with combination of IGB climate dataset to analyse the impact of SCA change on streamflows of each altitude zone. The observed hydro-climatic and IGB climatic dataset for the period of 2001 to 2010  is  referred as baseline (observed) and baseline (GCMs) for future climate change investigations for the different time lengths of 2030s (2030–2039), 2060s (2060–2069) and 2090s (2090–2099), as discussed in Sect. 5.2. The potential changes in climate variables (temperature and precipitation) and streamflows were assessed by taking average of four GCMs belongs to each of RCP8.5 and RCP4.5. The detail of climate change scenarios adopted in this study are given in following sections.





### 4.2.1 Based on RCPs scenarios (RCPs+BSCA)

Daily precipitation and mean air temperature were extracted form IGB climate dataset for a baseline period of 10 years (2001–2010) for each climatic stations located within the HRC. The large uncertainties were found when IGB climate dataset was compared with the observations. Therefore bias correction of IGB climatic data was done using the delta technique to derive

improved climatic data for further climate change analysis. The efficiency of delta approach has been discussed in detail by several researchers (Teutschbein and Seibert, 2012; Burhan et al., 2015; Maraun, 2016). The corrected IGB climate dataset during 2030s, 2060s and 2090s were used as an input in hydrological model to project the prospective streamflows in each altitude zone of HRC, for RCP8.5 and RCP4.5 scenarios by considering SCA unchanged i.e. baseline (observed) SCA (BSCA). Further, the anticipated changes in streamflows of each altitude zone of HRC were evaluated in comparison with baseline

(observed) streamflows.

### 4.2.2 Based on hypothetical scenarios (RCPs-HSCA)

Further, the study was also carried out to evaluate the impact of hypothetical change in SCA (HSCA) on streamflows by considering percent change (decrease) in SCA (-5%HSCA for 2030s, -10%HSCA for 2060s and -15%HSCA for 2090s) corrected RCPs climate dataset denoted as (RCPs-%HSCA). The hypothetical changes in SCA (HSCA) were assumed due to

15 unavailability of change in SCA under future scenarios. Because the HRC is dominantly covered by snow mostly during winter season (76 %, see Fig. 4), therefore, it is essential to investigate the sensitivity of HSCA on streamflows of HRC.

### 4.2.3 Bias correction

Delta technique was used to correct the projected precipitation and temperature data using observed and GCM baseline data from 2001 to 2010. The equations used in delta method are described below.

$$V_{tuned} = \frac{\overline{V_{obs}}}{\overline{V_{ref}}} \qquad (2)$$

$$S_{tuned} = \frac{\overline{S_{obs}}}{\overline{S_{ref}}} \qquad (3)$$

$$E_s = \left(V_{proj} - \overline{V_{ref}}\right) . S_{tuned} \qquad (4)$$

$$E_{proj} = E_s + \left(\overline{V_{ref}} . V_{tuned}\right) \qquad (5)$$

$\overline{V_{obs}}$ is the observed climatology, $\overline{V_{ref}}$ is the reference climatology for the GCM/RCM baseline, $V_{tuned}$ is the adjusted factor for

mean climate, $\overline{S_{obs}}$ is the standard deviation of monthly observed data set, $\overline{S_{ref}}$ is the standard deviation of GCM/RCM, $S_{tuned}$ is the signal to noise ratio, $V_{proj}$ is the particular projected month that needs correction, $E_s$ is the signal enhance or signal dampened for particular projection month and $E_{proj}$ is the bias corrected climatic variable for particular month (Maraun, 2016; Burhan et al., 2015; Hay et al., 2000).



## 5 Results and discussion

### 5.1 Application of SRM

The optimized parametric values during calibration and validation are described in Tables 2. SRM offers to calibrate and validate parametric values spatially (zone wise) and temporally (season wise). The parametric values of snowmelt runoff and rainfall runoff coefficients, $C_S$ and $C_R$, were found between 0.25-0.5 and 0-0.4, respectively. Correspondingly, the lapse rate, DDF, and lag time were found between 0.45-0.7 °C 100 m$^{-1}$, 0.4-0.6 cm °C-d$^{-1}$, and 6-18 h, respectively.

**Table 2: Calibrated parametric values for SRM model for HRC.**

The simulation efficiency of SRM was described by two statistical parameters, coefficient of determination ($R^2$) and Nash Sutcliff coefficient (NS). The statistical parameters for calibration (2001–2006) and validation (2008–2010) on annual and seasonal basis are given in Table 3. The statistical parameter value for NS coefficient and $R^2$ during calibration (validation) period were 0.92 (0.89) and 0.95 (0.92), respectively. Overall the SRM perform very well during all seasons (pre-monsoon, winter and monsoon) but it is observed that SRM performance during winter and pre-monsoon seasons is significantly higher than monsoon season as it is stated by Azmat et al. (2016a) that SRM is rather less efficient during rainfall season as compared with snow accumulation and melting seasons. During winter season, the $R^2$ and NS coefficient over the calibration (validation) period ranges from 0.97 (0.93) to 0.93 (0.89), respectively. Similarly, during monsoon season, the $R^2$ and NS coefficient varies between 0.73 to 0.75 and 0.69 to 0.71, respectively. Overall it is noted that SRM is fairly well efficient in high-altitude glaciers and snow-fed HRC.

**Table 3: Performance parameters during calibration (2001–2006) and validation (2008–2010) periods of SRM for HRC.**

The SRM was also run for 9 years (2001–2010 excluding 2007) period and scattered plots were drawn between observed streamflows and simulated streamflows during 9–year simulation period as shown in Fig. 5a. Over 9–years simulation, the $R^2$ (NS) coefficient values was remain 0.96 (0.91). Similarly, the efficiency of SRM was found higher in winter season (b) in contrast with pre-monsoon (c), monsoon (d) seasons. Overall, it is noticed that the SRM reproduce streamflows very efficiently during all seasons (winter, pre-monsoon and monsoon).

**Figure 5a. Scatter plots for seasonal performance of SRM during 9-years simulation (a) Annual, (b) Winter, (c) Pre-monsoon and (d) Monsoon.**

A minor disagreement between results obtained by previous studies were found which may possibly be due to the difference in study area characteristics i.e. rainfall dominant catchment and catchment topography. For example, Azmat et al. (2015) applied SRM in Jhelum River catchment and stated that the SRM is incapable to generate the rapid peaks of rainfall-runoff, a



significant less efficiency was found over rapid fluxes of peak streamflows. This disagreement is may be associated as the Jhelum River catchment is partially subjective to seasonal SCA and the major portion of the catchment is biased by the monsoon rainfall-runoff and HRC is partially subjective to the rainfall-runoff and mainly by the SCA. Although, the differences in altitude largely affects the precipitation and temperature which are the basic input variables to generate runoff volume (Kult et al., 2014).

The season wise simulated results for each zone are presented in Fig. 5b and percent contribution of runoff generated by each zone in described in Table 4a.

**Figure 5b: Zone wise seasonal contribution of streamflows during 9-years simulation period.**

The zone-wise contribution chart depicted that Zone(3) and (4) which have altitude ranges as 3500–4500 m a.s.l and 4500–5500 m a.s.l. are the major contributor toward the streamflows during 2001 to 2010 (except 2007), respectively and Zone(1), (2) and (5) are participating very less by 3.6 % (9.6 $m^3 s^{-1}$), 8.9 % (23.9 $m^3 s^{-1}$), and 3.2 % (8.5 $m^3 s^{-1}$) of annual streamflows of 269 $m^3 s^{-1}$, respectively. Interestingly, Zone(6) is participating almost zero percent (0.1 $m^3 s^{-1}$) into annual streamflows generation. Overall, Zone(4) is contributing almost half (49 %) of the annual streamflows which is 131.7 $m^3 s^{-1}$. It is also noted that in pre-monsoon and winter seasons, Zone(3) is contributing more streamflows (65.4 % and 32.9 %, respectively) than Zone(4). But in case of Monsoon season the relation is totally inverse in which 64.8 % (441.2 $m^3 s^{-1}$) is contributed by Zone(4) of total monsoon streamflows of 680.7 $m^3 s^{-1}$.

**Table 4a: Zone wise seasonal percent contribution during 9-years simulation period for HRC.**

The streamflows were also calculated per 100 $km^2$ area for each zone since each zone has different area. So it was necessary to evaluate the streamflows volume produced per unit area. After evaluation, the streamflows volume were ranked zone wise as Zone(3), (1), (4), (2), (5) and (6) from higher contribution to the lower contribution (Table 4b) on annual basis as 2.4 , 2.3, 2.2, 1.5, 0.6 and almost 0 $m^3 s^{-1}$ per 100 $km^2$ area, respectively. On seasonal basis, Zone(4) and (3) are producing more streamflows volume in monsoon season as 7.7 and 4.5 $m^3 s^{-1}$ per 100 $km^2$ area, respectively. Almost Zone(1), (3) and (4) are producing equal streamflows volume on annual basis per unit area.

Further, the SRM is being applied to evaluate the climate change impacts on streamflows in HRC, as discussed in Sect. 5.2.

**Table 4b: Zone wise seasonal specific flows ($m^3 s^{-1} km^{-2}$) during 9-years simulation period for HRC.**



### 5.2 Climate change impact assessment

### 5.2.1 Climatic projections

The IGB climate dataset is indicating an increasing trend in both precipitation and temperature on seasonal and annual basis during 2030s, 2060s and 2090s at all four climate stations as comparison with observed climate (2001–2010). Basin wide

variation in temperature (increase or decrease) and precipitation are given in Table 5. The station-wise increase in annual temperature by 0.7 (0.5), 0.5 (0.5), 0.8 (0.7), 0.6 (0.5) and 0.7 (0.6) ℃ was found for RCP8.5 (RCP4.5), respectively at Hunza , Naltar, Ziarat and Khunjrab climate stations in 2030s. It is observed that annual increase in temperature is more for RCP8.5 than RCP4.5 because of the high energy radiative scenario as it is also confirmed by Ali et al. (2015),  Burhan et al. (2015) and IPCC (2014). The maximum increase in temperature was found by 4.9 (2.0) ℃ at Ziarat climate station and minimum

temperature increase was found at Khunjrab climate station by 4.3 (1.8) ℃ under RCP8.5 (RCP4.5) scenario in 2090s. Rise in temperature on basin scale found that it will be increased by 0.7 (0.6), 2.4 (1.3) and 4.6 (1.9) ℃ under RCP8.5 (RCP4.5) scenario during 2030s, 2060s and 2090s, respectively. The basin-wise maximum increase in temperature is happening in pre-monsoon (monsoon) season by 5.5 (2.4) ℃ in 2090s under RCP8.5 (RCP4.5) scenario.

In case of precipitation, increasing trend was found at all climate stations for both RCPs. Though, maximum precipitation rise

was found during 2090s as given in Table 5. Annual maximum increase in precipitation is expected at Khunjrab climate station by 58.4 (55.2), 61.7 (71.2) and 94.4 (62.2) mm under RCP8.5 (RCP4.5) by 2030s, 2060s and 2090s, respectively. This rising trend at Khunjrab station is also well-defined by some researchers and acknowledged that the high-altitude range of the Karakorum is more vigorous hydrological zone and the altitude greater than 3500 m a.s.l. will receives maximum precipitation in coming future (Butz and Hewitt, 1986; Mukhopadhyay and Khan, 2014). At all climate station, RCP8.5 scenario is showing

ascending increase in precipitation from 2030s to 2090s from 29.9–60.7 mm, 5.3–40.7 mm, 26.9–52.4 mm, 58.4–94.4 mm at Hunza, Naltar, Ziarat and Khunjrab climate stations, respectively.  Basin scale investigation on precipitation increase showed that precipitation will increased by 30.1 (40.1), 35.2 (38.2), 63.3 (33.6) mm under RCP8.5 (RCP4.5) scenario during 2030s, 2060s and 2090s, respectively and maximum increase in precipitation is observed by 36.2 (27.8) mm in winter season during 2090s (2060s) under RCP8.5 (RCP4.5) scenario. Overall, winter season is showing more precipitation trends in both RCP

scenarios.

**Table 5: Projected temperature (℃) and precipitation (mm) deviations from observed climate during 2030s, 2060s and 2090s under RCP8.5 and RCP4.5 for HRC.**

### 5.2.2 Projected Streamflows

Projected streamflows during 2030s, 2060s and 2090s of HRC are evaluated by means of SRM using both climate change scenarios (RCP8.5 and RCP4.5) also in combination with hypothetical change in snow cover area (HSCA) scenarios which are further called as  RCPs+BSCA (Baseline Snow Cover Area) and RCPs-%HSCA, respectively as described in Sect 4.2. The observed data from 2001 to 2010 (excluding 2007) is referred as observed baseline for streamflows as well as for SCA.





a. Based on RCPs Scenarios (RCPs+BSCA)

The daily projected streamflows of HRC were generated by means of optimized parameters of SRM, whereas, input climate variables (precipitation and temperature) were substituted with RCPs bias corrected climate dataset while considering daily SCA unchanged i.e. Baseline SCA (BSCA). It is observed that projected streamflows are expected to increase under both RCPs scenarios as discussed in previous Sect. 5.2.1 that precipitation and temperature both are going to increase during 2030s, 2060s and 2090s (Table 6a).

The significant increase in streamflows was observed during 2060s with values of 92 % (233 $m^3 s^{-1}$), 53 % (361 $m^3 s^{-1}$) and 12 % (8.5 $m^3 s^{-1}$) for RCP8.5 and 28 % (71 $m^3 s^{-1}$), 39 % (266 $m^3 s^{-1}$) and 3 % (2 $m^3 s^{-1}$) for RCP4.5 during pre-monsoon, monsoon and winter seasons, respectively.

**Table 6a: Seasonal projected streamflows deviations (%) during 2030s, 2060s and 2090s for RCP8.5 and RCP4.5 under RCPs scenario (RCPs+BSCA).**

It is also seen from Table 6a that RCP8.5 scenario is showing maximum streamflows rising trend during pre-monsoon season and RCP4.5 is showing during monsoon season during 2030s, 2060s and 2090s. This seasonal maximum increasing trend discrimination in both scenarios is due to the maximum temperature rise during pre-monsoon season as 1.1, 3.2 and 5.5 ℃ for RCP8.5 and 1.4, 2.3 and 2.4 ℃ in monsoon season for RCP4.5 during 2030s, 2060s and 2090s, respectively (see Table 5). As it is discussed by several researchers that SRM is highly sensitive to temperature and SCA and less to precipitation so, it is generating more runoff (Dou et al., 2011; Kult et al., 2014; Azmat et al., 2016a). Zone wise contribution of streamflows are shown in Fig. 6 and percentage contribution of each zone is discussed in Table 6b.

**Figure 6: Zone-wise seasonal projected streamflows during 2030s, 2060s and 2090s for RCP8.5 and RCP4.5 under RCPs scenario (RCPs+BSCA).**

Annually, 48.7 (48.1), 50.4 (49.3) and 50.9 (50.7) % streamflows is contributed by Zone (4) and 37.4 (37.2), 32.7 (33.7) and 30 (31.2) % is contributed by Zone(3) in annual streamflows of 312 (304), 422.3 (352.4) and 573 (384.7) under RCP8.5 (RCP4.5) during 2030s, 2060s and 2090s, respectively that is greater than 80 % of the annual streamflow. All altitude zones are showing increasing streamflows contribution toward the projected climate but altitude range 4500–5500 m a.s.l is highly responsible to generate streamflows (more than 50 %) than other altitude ranges and contribution of altitude range 3500–4500 m a.s.l is also more than 30 % of the total streamflows of HRC. In monsoon season, it can be seen that Zone (4) streamflows are increasing from 441 $m^3 s^{-1}$ to 498.2 (489.3), 635.2 (578.4) and 789.9 (604.6) $m^3 s^{-1}$ for RCP8.5 (RCP4.5) during 2030s, 2060s and 2090s, respectively.





The major reason in producing these high streamflows is the limitation of the SCA which is considered as a constant for future streamflows projections that is unrealistic to the changing climate. To minimize this issue hypothetical change in SCA was considered for projected climate and discussed in next section.

**Table 6b: Zone-wise seasonal projected streamflows contribution (%) during 2030s, 2060s and 2090s for RCP8.5 and RCP4.5 under RCPs scenario (RCPs+BSCA)**

b. Based on Hypothetical Scenarios (RCPs-%HSCA)

Since, the HRC is dominantly covered with snow mostly during winter season (almost 76 %). So, the study was conducted to investigate the impact of change in SCA on streamflows using RCPs climate dataset in combination with percent change (decrease) in SCA (-5%HSCA for 2030s, -10%HSCA for 2060s and -15%HSCA for 2090s) that is denoted as (RCPs-%HSCA) and discussed in Table 7a.

**Table 7a: Seasonal projected streamflows deviations (%) during 2030s, 2060s and 2090s for RCP8.5 and RCP4.5 under RCPs scenario (RCPs-%HSCA).**

(i)   For RCPs-5%HSCA scenario, the SCA assumed be decreased by 5 % by 2030s for both RCP8.5 and RCP4.5. Decreasing 5 % SCA resulting decrease in mean annual streamflows by 2 % (5 $m^3 s^{-1}$) and 4 % (12 $m^3 s^{-1}$), for RCP8.5 and RCP4.5, respectively.

(ii)   For RCPs-10%HSCA scenario, with SCA decreased by 10 % during 2060s for both RCP8.5 and RCP4.5, resulting an increase in streamflows approximately by 18 % for RCP8.5 and decreased in streamflows by 2 % for RCP4.5 was found. That means increase in temperature (2.4 ℃) under RCP8.5 is enough to generate more streamflows under less SCA. While RCP4.5 (Temperature increase by 1.3 ℃) is not able to generate streamflows equal to annual baseline streamflows.

(iii)   For RCPs-15%HSCA scenarios, with SCA assumed to be decreased by 15 % for both RCPs, resulting in increase of 42 % and decreased of 7 %, annual streamflows for RCP8.5 and RCP4.5 scenarios, respectively. That means 15 % decrease in SCA for RCP8.5 can still generate more annual streamflows with increase in 4.6 ℃ temperature.

Overall the seasonal streamflows during pre-monsoon (monsoon) seasons are expected to increase from 253 (681) $m^3 s^{-1}$ to 261 (690), 367 (778), 509 (851) $m^3 s^{-1}$ by 2030s, 2060s and 2090s for RCP8.5, respectively. In the case of RCP4.5 streamflows are showing decreasing trend over all seasons except monsoon season of 2030s and 2060s, and pre-monsoon season of 2090s. Fig. 7 and Table 7b are showing the zone wise seasonal streamflows contribution and it can be seen that zone wise contribution under both RCPs during pre-monsoon and monsoon seasons are showing increasing trend for Zone(4).   The streamflows



increment is considerably high under RCP8.5 scenario in comparison of RCP4.5 which in corresponding is showing a decrease in streamflows or almost equal to baseline (observed) streamflows (Table 7a).

Further, minor influence of change in SCA on streamflows under RCP8.5-15%HSCA (2090s) scenario was observed that the projected streamflows for decrease in SCA are still greater than that of baseline. This fact is associated with the significant rise in temperature during 2060s and 2090s as compared to 2030s (see Table 7a for RCP8.5) which can even produce more streamflows. It should be noted that with reduction in SCA under RCP4.5, the streamflows are expected to approach baseline (observed) which is also associated with the aforementioned fact of temperature rise.

**Figure 7: Seasonal projected streamflows during 2030s, 2060s and 2090s for RCP8.5 and RCP4.5 under RCPs scenario (RCPs-%HSCA).**

Moreover, an increase in mean annual streamflows by 23 % (62 $m^3 s^{-1}$) is approximated by the increase of 1 °C mean annual temperature. Previously, Tahir et al. (2011b) specified that an increase of 33 % streamflows whereas, Akhtar et al. (2008 and Archer (2003) and stated 16 % increase in streamflows in HRC by rise of 1 °C mean temperature. Similarly, Forsythe et al. (2012) stated that ±2ºC change in temperature will result in ±20 % change in mean summer runoff in Upper Indus basin. This disagreement may associated with different data, methodologies, hypothesis and limitations under which current study is conducted in comparison with previous studies that used hypothetical climate change scenarios for (linear increase in temperature and precipitation).

**Table 7b: Seasonal projected streamflows contribution (%) during 2030s, 2060s and 2090s for RCP8.5 and RCP4.5 under RCPs scenario (RCPs-%HSCA).**

Overall, the variation in temperature and SCA are more dominant for the HRC production of streamflows under current and future climate regards as stated by the Kult et al. (2014) that the SRM sensitivity to the temperature and SCA. Therefore, it is important to build large water storage reservoirs to manage and deal with droughts and floods conditions consequently that will happened from the climate change.

## 6 Conclusions

This study used a snowmelt-runoff model (SRM) for Hunza River catchment (HRC) to simulate zone wise daily streamflows. Sensitivity analysis was done on the basis of altitude zones and contribution of each altitude zone in streamflows that showed that altitude Zone(4) having altitude range 4500–5500 m a.s.l, is highly responsible that is producing almost half (131.8 $m^3 s^{-1}$) of the mean annual streamflows during 9 years simulation period. Climate projection datasets comprises of RCP8.5 and RCP4.5 scenarios were used to analyse the projected change in climate variables. The basin-wide mean annual temperature is projected to increase by 0.7 (0.6), 2.4 (1.3) and 4.6 (1.9) °C during 2030s, 2060s and 2090s under RCP8.5 (RCP4.5),





respectively. Similarly, precipitation is going to increase by 63.3 (33.6) mm for RCP8.5 (RCP4.5) during 2090s as compared to the current climate. A significant increasing trend is found in projected streamflows under Baseline SCA scenario (RCPs+BSCA) for both RCPs but in the case of Hypothetical change in SCA scenarios (RCPs-%HSCA) both RCPs are showing different results i.e. RCP8.5 is showing slightly increasing trend but RCP4.5 is showing slightly decreasing trend.

Over all RCP8.5 is showing more projected streamflows than RCP4.5 because of its high energy radiative phenomenon. The Zone(4) and Zone(3) have significant importance in producing streamflows that are contributing more than 80 % of total streamflow of Hunza River which are located in the mid altitude mountainous ranges (3500–5500 m a.s.l). Seasonally, in pre-monsoon and monsoon seasons increasing trend was observed for Zone(4) as compared with baseline streamflows. The sensitivity of HRC to temperature was also evaluated which described that 1 ºC increase in temperature will cause about 23 %

increase in mean annual streamflows while considering the SCA unchanged.

This study was conducted under the limitation of hypothetical projected SCA which suggests that water resources management should be executed to changing climate pattern in high mountainous Indus River Basin. Current and near future climate pattern is favourable for Indus River regarding high water flow which provide flexibility while distributing water to different sectors at mid/downstream regions. However, future water flow pattern is declining because of disappearance of snow and glaciers

melt area over the region which correspondingly means that mid/ downstream water allocation will be effected or reduced at some extent at same time when its requirement will be high to satisfy the future demands in different sectors e.g. agriculture, industrial and environment etc. If proper adaptations or managements will not be executed then we should be ready for upcoming future changes. A detail study is needed to be conducted on the basis of projected SCA to clearly address the water resources availability in this region.

*Data availability.* The data that support the findings of this study are available from the corresponding author upon request.

*Competing interests.* The authors declare that they have no conflict of interest.

*Acknowledgements.* Authors want to acknowledge the HI–AWARE consortium for the provision of fine resolution IGB projected climate dataset. This work is supported by the Strategic Priority Research Program of the Chinese Academy of

25 Sciences (Grant No. XDA20100104), the National Natural Science Foundation of China (Grant No. 41630856), the 13th Five-year Informatization Plan of Chinese Academy of Sciences (Grant No. XXH13505-06) and the International Partnership Program of Chinese Academy of Sciences (Grant No. 131C11KYSB20160061).



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





**Fig. 1a**



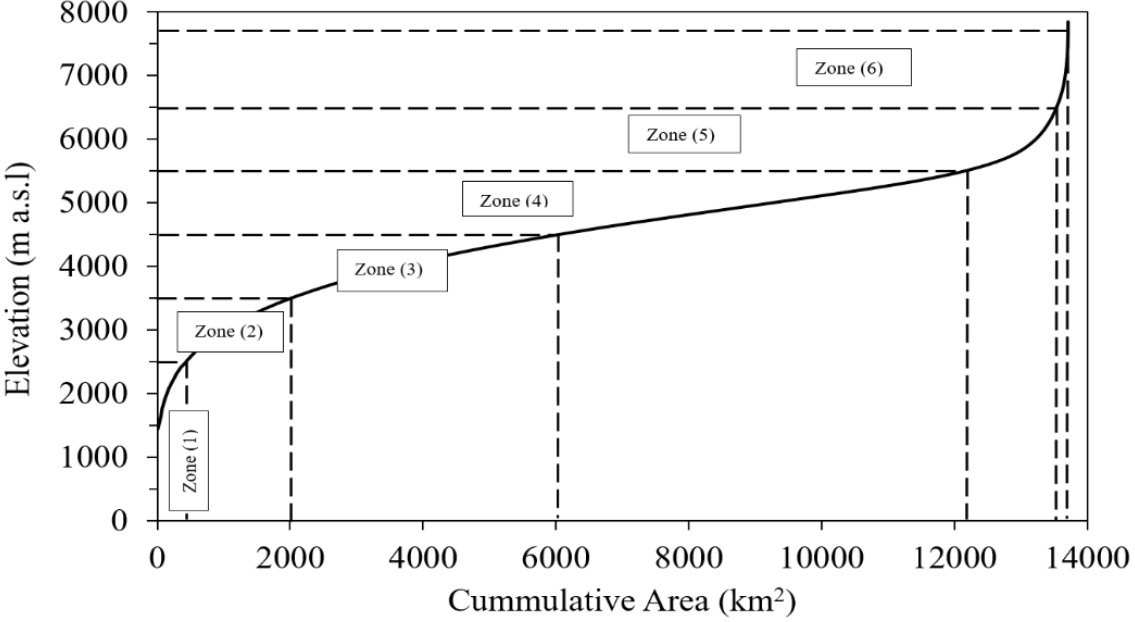

**Fig. 1b**

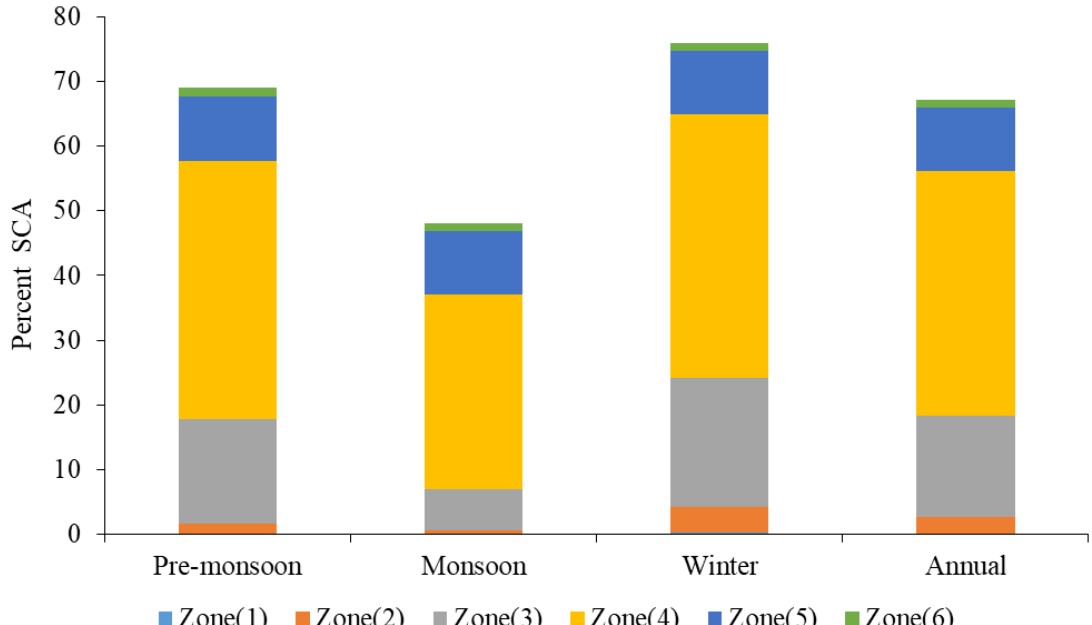

**Fig. 2**



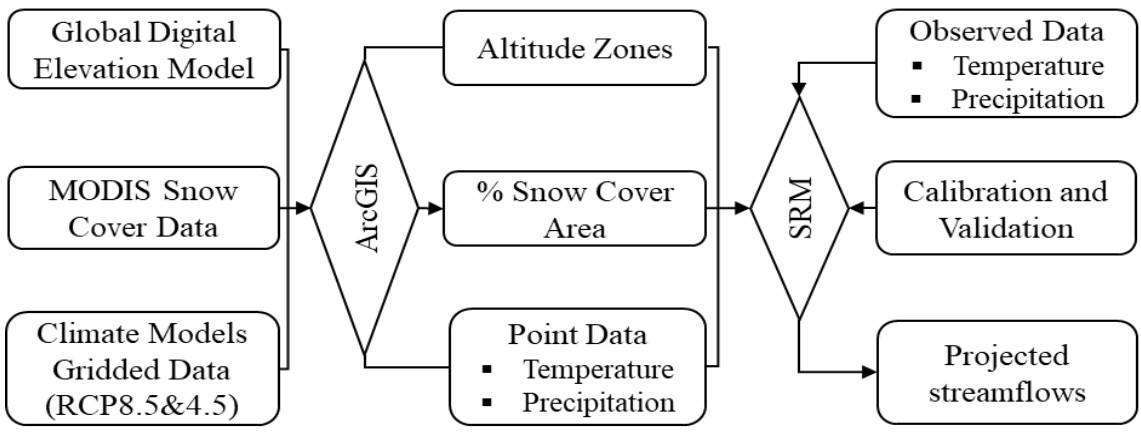

**Fig. 3**

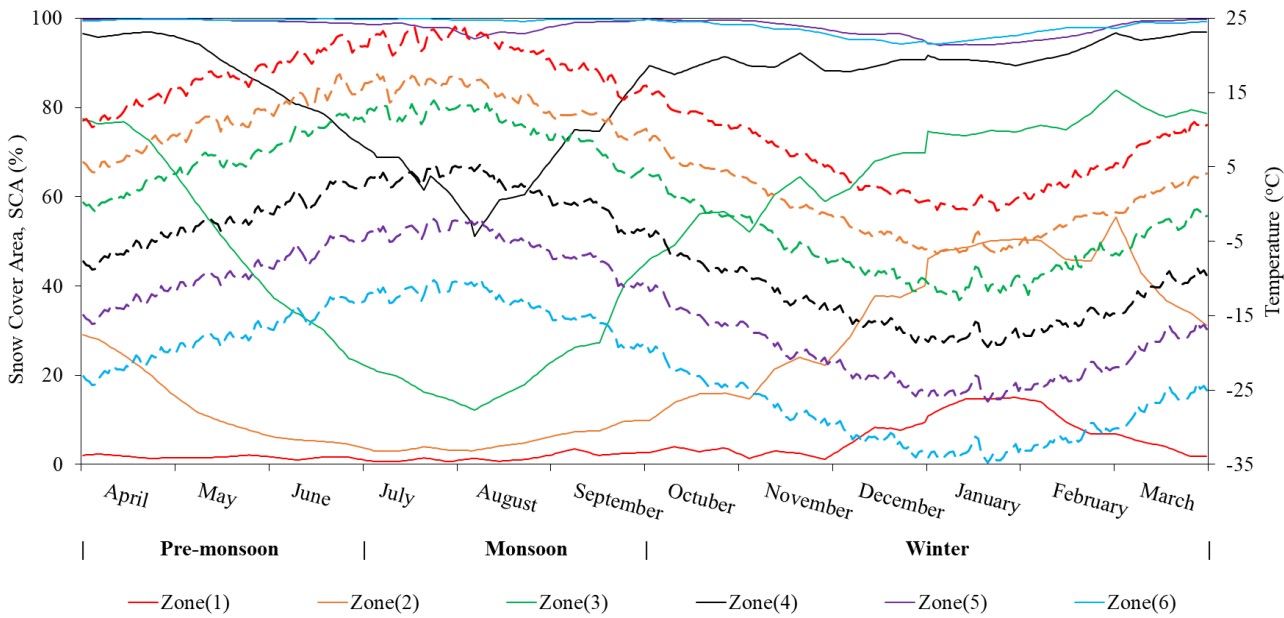

**Fig. 4**

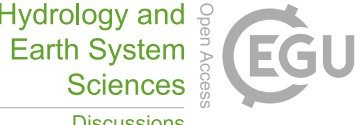

**Fig. 5a**



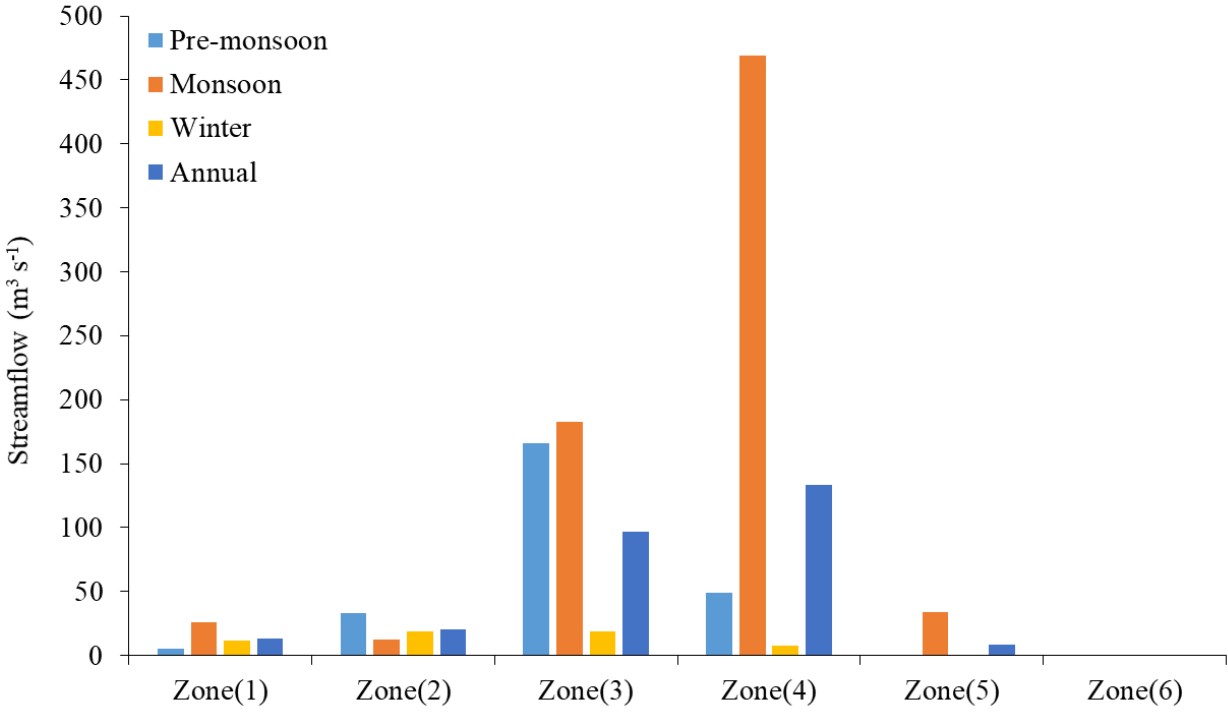

**Fig. 5b**



**Fig. 6**





**Fig. 7**



**Table 1**

| Altitude Zone | Altitude range (m a.s.l.) | Mean elevation (m a.s.l.) | Area (km²) | Area (%) | Max SCA (%) | Min SCA (%) | Climate stations |
|---|---|---|---|---|---|---|---|
| Zone(1) | 1395–2500 | 1965 | 431 | 3.1 | 7 | 1 | Hunza |
| Zone(2) | 2500–3500 | 3000 | 1581 | 11.5 | 54 | 3 | Naltar |
| Zone(3) | 3500–4500 | 4000 | 4025 | 29.3 | 83 | 13 | Ziarat |
| Zone(4) | 4500–5500 | 5000 | 6127 | 44.7 | 96 | 52 | Khunjrab |
| Zone(5) | 5500–6500 | 6000 | 1377 | 10.0 | 98 | 96 | − |
| Zone(6) | 6500–7849 | 7175 | 177 | 1.3 | 98 | 97 | − |

**Table 2**

| Parameters | Zone Wise Parametric Values | | | | | |
|---|---|---|---|---|---|---|
| | Zone(1) | Zone(2) | Zone(3) | Zone(4) | Zone(5) | Zone(6) |
| Lapse Rate (°C 100 m⁻¹) | 0.45 | 0.5 | 0.55 | 0.6 | 0.7 | 0.7 |
| $T_{crit}$ (°C) | 1 | 1 | 1 | 1 | 1 | 1 |
| DDF (cm °C-d⁻¹) | 0.4 | 0.45 | 0.5 | 0.55 | 0.55 | 0.6 |
| Lag Time (h) | 6 | 6 | 12 | 12 | 18 | 18 |
| $C_S$ | 0.3 | Jun–Aug = 0.35 Sep–May = 0.3 | Jun–Aug = 0.35 Sep–May = 0.3 | Jun–Aug =0.4 Sep–May = 0.3 | Jun–Aug = 0.5 Sep–May = 0.3 | Jun–Aug = 0.4 Sep–May = 0.25 |
| $C_R$ | Jun–Aug = 0.4 Sep–May = 0.25 | Jun–Aug = 0.35 Sep–May = 0.3 | Jun–Aug = 0.3 Sep–May = 0.25 | Jun–Aug = 0.25 Sep–May = 0.2 | Jun–Aug = 0 Sep–May = 0.15 | 0 |
| RCA | 1 | 1 | 1 | Jun–Aug = 1 Sep–May = 0 | Jun–Aug = 1 Sep–May = 0 | 0 |
| Xc | 1.08 | 1.08 | 1.08 | 1.08 | 1.08 | 1.08 |
| Yc | 0.022 | 0.022 | 0.022 | 0.022 | 0.022 | 0.022 |

$T_{crit}$= critical temperature; DDF= degree day factor; RCA= rainfall contributing area; $C_S$ snowmelt runoff coefficient; $C_{R=}$ runoff coefficient; $X_C$ and $Y_C$ = recession coefficients





**Table 3**

| Season | | Calibration (2001–2006) | Validation (2008–2010) |
|---|---|---|---|
| Annual (Jan–Dec) | | | |
| | $R^2$ | 0.95 | 0.92 |
| | NS Coefficient | 0.92 | 0.89 |
| Winter Season (Oct–March) | | | |
| | $R^2$ | 0.97 | 0.93 |
| | NS Coefficient | 0.93 | 0.89 |
| Pre-monsoon (April–June) | | | |
| | $R^2$ | 0.85 | 0.81 |
| | NS Coefficient | 0.82 | 0.79 |
| Monsoon (July–Sep) | | | |
| | $R^2$ | 0.75 | 0.73 |
| | NS Coefficient | 0.71 | 0.69 |

**Table 4a**

| Season | Zone(1) | Zone(2) | Zone(3) | Zone(4) | Zone(5) | Zone(6) |
|---|---|---|---|---|---|---|
| Pre-monsoon | 2.1 | 13.0 | 65.4 | 19.4 | 0.1 | 0.0 |
| Monsoon | 1.5 | 3.7 | 25.3 | 64.8 | 4.7 | 0.0 |
| Winter | 20.1 | 32.7 | 32.9 | 13.9 | 0.4 | 0.0 |
| Annual | 3.6 | 8.9 | 35.4 | 49.0 | 3.2 | 0.0 |

**Table 4b**

| Season | Zone(1) | Zone(2) | Zone(3) | Zone(4) | Zone(5) | Zone(6) |
|---|---|---|---|---|---|---|
| Pre-monsoon | 1.3 | 2.1 | 4.1 | 0.8 | 0.0 | 0.0 |
| Monsoon | 2.5 | 1.7 | 4.5 | 7.7 | 2.5 | 0.2 |
| Winter | 2.7 | 1.2 | 0.5 | 0.1 | 0.0 | 0.0 |
| Annual | 2.3 | 1.5 | 2.4 | 2.2 | 0.6 | 0.0 |





**Table 5**

| Decade | Season | Temperature (°C) | | Precipitation (mm) | |
|---|---|---|---|---|---|
| | | RCP8.5 | RCP4.5 | RCP8.5 | RCP4.5 |
| 2030s | Pre-Monsoon | 0.9 | 0.5 | 7.0 | 11.4 |
| | Monsoon | 1.1 | 1.4 | 14.4 | 16.1 |
| | Winter | 0.3 | 0.2 | 19.1 | 19.4 |
| | Annual | 0.7 | 0.6 | 30.1 | 40.1 |
| 2060s | Pre-Monsoon | 3.2 | 1.2 | 7.0 | 4.8 |
| | Monsoon | 2.7 | 2.3 | 17.1 | 7.7 |
| | Winter | 1.9 | 0.8 | 22.4 | 27.8 |
| | Annual | 2.4 | 1.3 | 35.2 | 38.2 |
| 2090s | Pre-Monsoon | 5.5 | 2.1 | 11.1 | 2.7 |
| | Monsoon | 4.7 | 2.4 | 23.5 | 7.8 |
| | Winter | 4.1 | 1.5 | 36.2 | 23.2 |
| | Annual | 4.6 | 1.9 | 63.3 | 33.6 |

**Table 6a**

| Decade | Season | RCP8.5 | RCP4.5 |
|---|---|---|---|
| 2030s | Pre-Monsoon | 21 | 7 |
| | Monsoon | 20 | 18 |
| | Winter | 7 | 1 |
| | Annual | 16 | 13 |
| 2060s | Pre-Monsoon | 92 | 28 |
| | Monsoon | 53 | 39 |
| | Winter | 12 | 3 |
| | Annual | 57 | 31 |
| 2090s | Pre-Monsoon | 193 | 66 |
| | Monsoon | 90 | 43 |
| | Winter | 79 | 5 |
| | Annual | 113 | 43 |



**Table 6b**

| Season | Zone(1) | Zone(2) | Zone(3) | Zone(4) | Zone(5) | Zone(6) |
|---|---|---|---|---|---|---|
| | RCP8.5+BSCA (2030s) | | | | | |
| Pre-monsoon | 5.2 | 5.7 | 60.1 | 28.8 | 0.2 | 0.0 |
| Monsoon | 1.1 | 2.8 | 28.8 | 61.0 | 6.3 | 0.1 |
| Winter | 27.8 | 18.3 | 38.0 | 15.5 | 0.5 | 0.0 |
| Annual | 4.6 | 5.0 | 37.4 | 48.7 | 4.3 | 0.0 |
| | RCP8.5+BSCA (2060s) | | | | | |
| Pre-monsoon | 3.5 | 5.9 | 51.0 | 38.1 | 1.5 | 0.0 |
| Monsoon | 1.1 | 3.9 | 23.2 | 61.0 | 10.5 | 0.3 |
| Winter | 23.7 | 18.7 | 38.4 | 18.2 | 1.0 | 0.0 |
| Annual | 3.9 | 5.8 | 32.7 | 50.4 | 7.0 | 0.2 |
| | RCP8.5+BSCA (2090s) | | | | | |
| Pre-monsoon | 2.5 | 6.0 | 44.9 | 43.2 | 3.3 | 0.1 |
| Monsoon | 1.1 | 4.3 | 18.9 | 61.0 | 14.0 | 0.7 |
| Winter | 16.8 | 18.1 | 43.6 | 19.5 | 1.9 | 0.0 |
| Annual | 3.2 | 6.2 | 30.0 | 50.9 | 9.2 | 0.4 |
| | RCP4.5+BSCA (2030s) | | | | | |
| Pre-monsoon | 5.9 | 6.9 | 63.4 | 23.7 | 0.1 | 0.0 |
| Monsoon | 1.0 | 2.8 | 28.5 | 60.9 | 6.7 | 0.1 |
| Winter | 27.0 | 18.4 | 38.0 | 16.1 | 0.5 | 0.0 |
| Annual | 4.7 | 5.2 | 37.2 | 48.1 | 4.6 | 0.0 |
| | RCP4.5+BSCA (2060s) | | | | | |
| Pre-monsoon | 5.2 | 6.7 | 58.7 | 28.9 | 0.5 | 0.0 |
| Monsoon | 1.2 | 4.0 | 24.5 | 61.1 | 8.9 | 0.2 |
| Winter | 26.7 | 18.5 | 37.2 | 16.7 | 0.8 | 0.0 |
| Annual | 4.6 | 6.0 | 33.7 | 49.3 | 6.2 | 0.1 |
| | RCP4.5+BSCA (2090s) | | | | | |
| Pre-monsoon | 5.7 | 5.6 | 53.9 | 34.0 | 0.8 | 0.0 |
| Monsoon | 1.9 | 4.2 | 21.4 | 62.1 | 10.1 | 0.2 |
| Winter | 30.5 | 15.0 | 34.0 | 19.6 | 0.9 | 0.0 |
| Annual | 5.7 | 5.5 | 31.2 | 50.7 | 6.8 | 0.2 |





**Table 7a**

| Decade | Season | Hypothetical Scenarios (RCPs-%HSCA) | |
|---|---|---|---|
| | | RCP8.5 | RCP4.5 |
| | | -5%HSCA | |
| 2030s | Pre-Monsoon | 3 | -10 |
| | Monsoon | 1 | 2 |
| | Winter | -25 | -23 |
| | Annual | -2 | -4 |
| | | -10%HSCA | |
| 2060s | Pre-Monsoon | 45 | -4 |
| | Monsoon | 14 | 3 |
| | Winter | -12 | -22 |
| | Annual | 18 | -2 |
| | | -15%HSCA | |
| 2090s | Pre-Monsoon | 101 | 4 |
| | Monsoon | 25 | -7 |
| | Winter | 19 | -25 |
| | Annual | 42 | -7 |



**Table 7b**

|  | Zone(1) | Zone(2) | Zone(3) | Zone(4) | Zone(5) | Zone(6) |
|---|---|---|---|---|---|---|
| Season | RCP8.5-5%HSCA (2030s) | | | | | |
| Pre-monsoon | 5.8 | 3.9 | 59.4 | 30.6 | 0.2 | 0.0 |
| Monsoon | 0.8 | 0.1 | 26.0 | 66.0 | 7.0 | 0.1 |
| Winter | 27.8 | 10.1 | 43.6 | 17.9 | 0.6 | 0.0 |
| Annual | 4.7 | 2.1 | 36.0 | 52.4 | 4.7 | 0.1 |
| | RCP8.5-10%HSCA (2060s) | | | | | |
| Pre-monsoon | 4.1 | 2.8 | 49.7 | 41.7 | 1.7 | 0.0 |
| Monsoon | 0.7 | 0.9 | 17.3 | 68.2 | 12.6 | 0.4 |
| Winter | 23.6 | 9.8 | 42.1 | 23.0 | 1.4 | 0.0 |
| Annual | 3.9 | 2.4 | 29.1 | 56.1 | 8.3 | 0.2 |
| | RCP8.5-15%HSCA (2090s) | | | | | |
| Pre-monsoon | 3.0 | 2.3 | 42.6 | 48.1 | 3.9 | 0.1 |
| Monsoon | 0.6 | 0.9 | 10.0 | 69.8 | 17.8 | 0.8 |
| Winter | 17.5 | 7.7 | 45.8 | 26.2 | 2.7 | 0.1 |
| Annual | 3.3 | 2.1 | 24.8 | 57.8 | 11.5 | 0.5 |
| | RCP4.5-5%HSCA (2030s) | | | | | |
| Pre-monsoon | 6.6 | 5.0 | 63.1 | 25.3 | 0.1 | 0.0 |
| Monsoon | 0.8 | 0.1 | 25.7 | 65.9 | 7.4 | 0.1 |
| Winter | 26.9 | 10.0 | 43.7 | 18.6 | 0.6 | 0.0 |
| Annual | 4.8 | 2.2 | 35.9 | 51.9 | 5.1 | 0.0 |
| | RCP4.5-10%HSCA (2060s) | | | | | |
| Pre-monsoon | 6.2 | 3.4 | 57.7 | 32.1 | 0.6 | 0.0 |
| Monsoon | 0.8 | 1.0 | 18.3 | 68.9 | 10.8 | 0.2 |
| Winter | 26.7 | 10.1 | 40.6 | 21.5 | 1.1 | 0.0 |
| Annual | 4.7 | 2.5 | 29.7 | 55.4 | 7.4 | 0.2 |
| | RCP4.5-15%HSCA (2090s) | | | | | |
| Pre-monsoon | 8.2 | 2.0 | 52.0 | 38.8 | 1.0 | 0.0 |
| Monsoon | 1.9 | 0.7 | 11.9 | 72.8 | 13.1 | 0.3 |
| Winter | 35.8 | 5.8 | 35.7 | 27.1 | 1.4 | 0.0 |
| Annual | 7.1 | 1.4 | 24.9 | 59.0 | 8.7 | 0.2 |