# Peer review of "Climate change and runoff contribution by hydrological zones of cryosphere catchment of Indus River, Pakistan"

_Hydrology and Earth System Sciences, 2018_

## Referee Comment (RC1) · Anonymous Referee #1 · 20 Nov 2018

SRM is fundamentally a temperature index model, that is calibrated on the relationship between runoff and air temperature. While it may be the only type of model that can be applied to the Hindukush, due to limited data, it is - in my opinion - totally inappropriate to use this type of model for a climate scenario assessment. My recommendation is rejection. If the authors want to do this sort of assessment, they should use a physics based snow model, like CHRM, JULES, SnoPack (Alpine 3D), or perhaps iSnobal (though iSnobal currently does not have a hydrology or streamflow component). Addressing this sort of analysis with SRM shows an inherent lack of understanding about the relationship between climate, snow deposition and melt, and hydrology. The relationship between temperature and climate is not simple... Warming also changes

humidity, wind, and precipitation. The complexity of that interaction will change and alter the relationship between air temperature, snow deposition and melt. My greatest concern is that a paper like this gets published, resulting in quasi-legitimization of the approach, and then we have to fight that battle again and again. This approach is not appropriate for a journal like HESS.

---

## Referee Comment (RC2) · Anonymous Referee #2 · 10 Dec 2018

This paper addresses a very important topic and basin. The changing headwaters of the Indus are of major societal concern, because of downstream effects, and the science is challenging and poorly understood. However, the paper at its current stage of development is not yet publishable. It needs: A clearer review of the relevant international literature relating to Himalayan climate change and flow studies Justification for the hydrological model selected, including relevant comparable applications Modelling methodology to be more fully explained. How was the model optimized given limited data, what criteria were used, what temporal resolution. How were multiple elevation zones fitted without supporting flow data? The paper speculates about the future of snow covered area and uses this in simulations. However there is no explanation, discussion or justification given for the changes Future climate scenarios are reported to be wetter, but flows are said to decrease. How can this be? Finally, the paper is generally written at a level of English that is not suitable for publication. The authors should take advice on the English presentation in any further submission.

---

## Referee Comment (RC3) · Anonymous Referee #3 · 16 Dec 2018

The paper by Jamal and co-authors examines the influence of climate change on runoff and other hydrological processes in the mountains of the Hunza River Catchment. Two future climate change scenarios are used to credit the future of the basin using the Snowmelt-Runoff Model. At present, this manuscript should be rejected. The English is very poor, and requires many dozens of substantive changes to improve clarity and correctness. In terms of a scientific paper, it reads more like an engineering report than a contribution to the literature. There is a lack of appropriate referencing as there has been considerable work in this area (recently) that has been ignored. Furthermore, the SRM model is an antiquated snowmelt model that I do not believe is appropriate for climate change scenarios as it does not explicitly account for the energy balance of

the snowpack and other changes in surface-atmosphere exchanges when temperature changes. I fail to see the scientific contribution of this paper in its current form.
* * *

---

## Editor Comment (EC1) · DeBeer (Editor) · 19 Dec 2018

Dear Authors, Thank you for your submission to HESS and to this special issue. Unfortunately the reviewer comments indicate that the paper is not suitable for publication at this time, but they have given some advice on how to strengthen the study and improve the manuscript. Although this is beyond the scope of major revisions for this issue, I would encourage you to take up this advice for a potential future re-submission. The HESS editorial staff will now close the discussion and they ask that you please withdraw the paper after the discussion has been closed. Again, thank you and best wishes as you continue this work.

---

## Author Comment (AC1) · 6 Jan 2019

We would like to thank Refree#1 for his thoughtful and constructive comments about our manuscript. We would like to give our response in four steps.

1- I do believe that SRM model use air temperature to estimate melt, they do not account for the influence of topographic shading, slope or aspect on melt rate. 2- As you mentioned that this kind of model can me only used in Hindukush region. This study area is in Hindukush and Karakorum region (missing in the manuscript) with limited data available. 3- We applied this simple model because of limited data available. The annual precipitation is around 375mm without considering any losses and the annual

streamflow is around 700mm within the study area. How we can apply a complex model even observed data does not represent the true hydrology in this region. 4- On the basis of some previous studies we selected this model. The studies stated that comparisons of temperature index models with full energy balance models found the performance of simple model is far better that energy based complex model when accurate meteorological information is not available (Réveillet et al. 2018; Magnusson et al. 2015).

Réveillet M, Six D, Vincent C, Rabatel A, Dumont M, Lafaysse M, Morin S, Vionnet V, Litt M (2018) Relative performance of empirical and physical models in assessing the seasonal and annual glacier surface mass balance of Saint-Sorlin Glacier (French Alps). Cryosphere 12:1367–1386. https://doi.org/10.5194/tc-12-1367- 2018. Magnusson J,Wever N, Essery R, Helbig N, Winstral A, Jonas T (2015) Evaluating snow models with varying process representations for hydrological applications. Water Resour Res 51(4):2707–2723. https://doi.org/10.1002/2014WR016498

---

## Author Comment (AC2) · 6 Jan 2019

We are very thankful to Refree#2 for their thoughtful comments on our manuscript. It is obvious from the review that we have to explain and clarify the ideas in the manuscript. In this regard, we will provide a response to all the major and minor comments of the Refree#2. We will later address all the comments at the time of further submission.

---

## Author Comment (AC3) · 6 Jan 2019

We are very thankful to Refree#3 for their thoughtful comments on our manuscript. It is obvious from the review that we have to explain and clarify the ideas in the manuscript and to improve the language written. We will incorporate your comments to make it valuable for the scientific community. As of SRM model is concern we applied this simple model because of limited data availability. Some previous studies stated that comparisons of temperature index models with full energy balance models found the performance of simple model is far better that energy based complex model when accurate meteorological information is not available. Overall, we will provide a response

to all the major and minor comments of the Refree#3 and will later address all the comments at the time of further submission.

---

## Author Comment (AC4) · 6 Jan 2019

We are very thankful to Editor (DeBeer) for their final suggestions about our manuscript. We will try to incorporate the suggestions and comments of all the referees to make this manuscript valuable for the research community and will make assure the quality to justify the norms of the Journal while considering re-submission.
* * *